# Community participation in the collaborative governance of primary health care facilities, Uasin Gishu County, Kenya

**Jackline Sitienei**[1,2]*, **Lenore Manderson**[1,3], **Mabel Nangami**[2]

**1** School of Public Health, University of the Witwatersrand, Johannesburg, South Africa, **2** Health Policy and Management Department, School of Public Health, College of Health Sciences, Moi University, Eldoret, Kenya, **3** Department of Anthropology, School of Social Sciences, Monash University, Clayton, Australia

* jcsitienei@cartafrica.org, sitieneij@yahoo.com

## Abstract

### Introduction

Community participation in the governance of health services is an important component in engaging stakeholders (patients, public and partners) in decision-making and related activities in health care. Community participation is assumed to contribute to quality improvement and goal attainment but remains elusive. We examined the implementation of community participation, through collaborative governance in primary health care facilities in Uasin Gishu County, Western Kenya, under the policy of devolved governance of 2013.

### Methods

Utilizing a multiple case study methodology, five primary health care facilities were purposively selected. Study participants were individuals involved in the collaborative governance of primary health care facilities (from health service providers and community members), including in decision-making, management, oversight, service provision and problem solving. Data were collected through document review, key informant interviews and observations undertaken from 2017 to 2018. Audio recording, notetaking and a reflective journal aided data collection. Data were transcribed, cleaned, coded and analysed iteratively into emerging themes using a governance attributes framework.

### Findings

A total of 60 participants representing individual service providers and community members participated in interviews and observations. The minutes of all meetings of five primary health care facilities were reviewed for three years (2014–2016) and eight health facility committee meetings were observed. Findings indicate that in some cases, structures for collaborative community engagement exist but functioning is ineffective for a number of reasons. Health facility committee meetings were most frequent when there were project funds, with discussions focusing mainly on construction projects as opposed to the day-to-day functioning of the facility. Committee members with the strongest influence and power had

**Data Availability Statement:** Governance is a sensitive topic and qualitative data is difficult to de-identify the data completely. Data from this study can be availed through approving ethics committee

-Institutional ethics and research Committee (IREC) Moi teaching and referral Hospital (MTRH) and Moi University, College of health Sciences. To access the data one can, write to the board chair, IREC, P.O. Box 3-30100, Eldoret, Kenya. Email: irecmtrh@gmail.com or contact@irec.or.ke. Contact person is Ms Cathrine Okwiri.

**Funding:** This research was supported by CARTA. CARTA is jointly led by the African Population and Health Research Center and the University of the Witwatersrand and funded by the Carnegie Corporation of New York (Grant No–B 8606.R02), Sida (Grant No:54100113), the DELTAS Africa Initiative (Grant No: 107768/Z/15/Z) and Deutscher Akademischer Austauschdienst (DAAD). The DELTAS Africa Initiative is an independent funding scheme of the African Academy of Sciences (AAS)'s Alliance for Accelerating Excellence in Science in Africa (AESA) and supported by the New Partnership for Africa's Development Planning and Coordinating Agency (NEPAD Agency) with funding from the Welcome Trust (UK) and the UK government. Funding was also received from Future Health systems and DFID under Grant No. PO (5683).

**Competing interests:** The authors have declared that no competing interests exist.

political connections or were retired government workers. There were no formal mechanisms for stakeholder forums and how these worked were unclear. Drug stock outs, funding delays and unclear operational guidelines affected collaborative governance performance.

## Conclusion

Implementing collaborative governance effectively requires that the scope of focus for collaboration include both specific projects and the routine functioning of the primary health care facility by the health facility committee. In the study area, structures are required to manage effective stakeholder engagement.

## Introduction

Collaborative governance as practiced in the health sector brings different stakeholders together for consensus-oriented decision making to address issues that affect service provision, responsibility over deliberations, and accountability. The structures for collaborative governance are provided by governments that legitimize and encourage the active participation of citizens on an individual basis and as representatives of specific entities [1, 2]. Collaborative governance provides an opportunity to support community empowerment and to contribute to and influence the goals and practices of services. This includes enhancing the capacity of services and community members to improve overall health and quality of life, to nurture cordial relationships between health facility staff, and to increase the extent to which services are responsive to users and their communities [3–5]. Although these collaborations are intended to ultimately contribute to responsive health care, there is limited evidence and lack of clarity about operational structures and processes needed for effective implementation [7–12].

Governance is a key determinant of performance of health care facilities [1–3]. Governance refers to the complex mechanisms, processes and institutions which enable actors to interact at constitutional, collaborative and operational levels to articulate their interests, mediate differences, and exercise legal rights and obligations in order to influence outcomes of public policy [4–6]. Health institutions characterized with poor governance face performance challenges, including in relation to their capacity to address or satisfy stakeholders concerns [3, 7]. One way of overcoming these governance challenges is involving citizens in the provision of the services they utilize. Both governance and citizen engagement can be looked at in a multilevel perspective. Ambiola and colleagues apply one such framework for primary health care governance at operational, collective and constitutional governance levels depending on influence. This present article focuses on collective governance level [8]. Governance attributes include accountability, transparency, participation, consensus orientation, equity and inclusion, effectiveness, efficiency, intelligence and information and power [6, 9]. 'Good' or 'bad' governance is measured in terms of the extent to which these attributes are implemented [1, 10].

Health care services in Kenya are organized into six levels of care. Level one is the community, level two is the dispensaries, level 3 is the health centers, level four is the sub-county and county hospitals, level five is the county referral hospitals, and level six is the national referral hospitals. Levels one, two and three constitute primary health care services, with formal facility based health care services provided at levels two and three. These primary health care facilities constitute the majority of Kenya's health facilities that should be accessible to majority of the population and are essential to attain key health outcomes [11]. There has been global effort by various multinational and bilateral bodies since the Alma-Ata Declaration of 1978 on the

effective implementation of primary health care services [12, 13]. Despite this, researchers have demonstrated that many countries in Low and Middle Income Countries (LMIC), including Kenya, have weak primary health care systems [14].

In 2010, Kenya initiated governance reforms to support devolution, following a referendum to split power between national and county governments. Devolution is a form of decentralization whereby authority is restructured and there is co–responsibility between national and county governments [15, 16]. The aim in limiting the role of centralized government was to enhance oversight, accountability, promote individual freedom and public participation as aspects of collaborative governance [28–31]. Devolution began to be implemented in 2013, when the newly elected government took office following general elections.

The constitution has many provisions for community participation, including participation in the management and oversight of public facilities. Counties receive 15% of equalization funds of the countries' budgetary allocation to be distributed among the ten service departments at county level, including education, development, agriculture and health. The Department of Health receives funds directly or indirectly from Local Authority Transfer Funds (LATIF), Constituency Development Funds (CDF), Health Sector Service Funds (HSSF) and hospital management service funds. Engaging community members in governance structures, such as through health facility committees (boards) or other stakeholder fora, is set out in the constitution and is often a donor requirement, for example, to use Health Sector Service Fund (HSSF) [15, 17, 18].

The National Health Sector Strategic Plan II of 2014–2018, issued by the Ministry of Health, refers to the absence of accountability mechanisms on committees and facility boards, suggesting that there was already, at time of writing, a problem with planning mechanisms, transparency, monitoring, supervision and reporting by collaborators. Further, the document reported limited implementation of a rights-based approach in health service delivery. The plan emphasizes the need to establish mechanisms for collaboration with all health-related partners by holding stakeholder forums, creating committees and boards, and carrying out dialogue days with community members at facility level.

From the beginning of measures for devolution nationwide, all counties introduced policy documents, guidelines and proposed structures to implement collaborative governance. The Uasin Gishu Strategic Plan 2013–2018 emphasizes a rights-based approach and the need to ensure the inclusion of minority and marginalized groups in governance by conducting quarterly health stakeholder meetings and encouraging regular facility committee meetings [19]. In addition, the County Projects Implementation and Management Act of 2014 provided a legal framework to implement collaborative governance through health facility committees [19]. The document outlines the mechanisms for the establishment, administration and functioning of these committees. Under the act, the health facility committee should comprise of five to nine members whose roles and responsibilities are shown in Table 1. Community members from the catchment villages covered by a given health facility were to elect representatives, with the election process presided over by the public health officer from the facility and the area chief (the government administrator who handles community issues within a geographical boundary such as at the sub-county level).

The contribution of collaborative governance in terms of accountability, community participation and consensus orientation, transparency, equity and inclusion in the delivery of primary health care has not been fully realized in Uasin Gishu County. At the time of the study I report on here, the county department of health was unclear on the need for and contribution of collaborators, including the health facility committee members. A review of the literature reveals mixed findings on engaging communities in health, and in Kenya, few studies have been conducted. For example, McCollum and colleagues compared experiences of health

**Table 1. Positions and responsibilities of health facility committee.**

| HFC Member | Key Responsibilities |
|---|---|
| Chairperson | Chairs HFC meetings, accounts signatory |
| Vice-Chairperson | Responsibilities of chairperson in absentia. |
| Secretary | Facility in-charge, accounts signatory, organizes meetings, records minutes |
| Public Health Officer | Member |
| Treasurer | Accounts signatory |
| Chief | Ex-officio member |
| Member of County Assembly (MCA) | Ex-officio member |
| Ward Administrator | Ex-officio member and accounts signatory |
| Disabled Representative | Represents interests of people with disability |
| Women's Representative | Represents interests of women |
| Youth Representative | Represents interests of youths |

Source: Uasin Gishu County Gazette Supplement Bill, 2014, showing membership and different roles of Health Facility Committee members.

system governance under decentralization in Kenya and Indonesia, and found weak accountability structures, limitations in community engagement and technological capacity in both countries [20]. Indonesia devolved the health system 15 years earlier, but still experienced challenges similar to Kenya, suggesting that the Kenyan government re-examine health governance to avert prolonged health governance challenges. Published studies on the implementation of governance at primary health care level reveal evidence gaps in terms of roles of social accountability, public-private partnerships and intersectoral collaboration. In this article, therefore, drawing on data from Uasin Gishu County, we examined the implementation of collaborative governance at primary health care facilities, looking at the structures in place for such operations as selection processes, the demarcation and undertaking of roles and responsibilities, and the challenges faced after implantation of devolution in Kenya in 2013. We illustrate the potential of collaborative governance to provide information to enable the implementation of more inclusive, responsive and accountable governance systems and sustainable collaborative efforts by all stakeholders.

## Materials and methods

### Study design

The aim of the study on which we report was to examine the implementation of collaborative governance at primary health care facilities in Uasin Gishu County, as introduced with devolved governance in Kenya in 2013. A multiple case study methodology was used as a suitable approach to understanding governance and community participation in a real life setting [21, 22].

### Study setting

The study was conducted between April 2016 and December 2017 in Uasin Gishu County, one of 47 counties in Kenya. This county was selected purposively for its strategic location in providing health services in Western Kenya [23], in an area that is politically volatile. hot spot. The county hosts the second largest referral hospital in Western Kenya and the neighbouring countries like Uganda, Sudan, Burundi, located in a metropolitan region, but serving both urban and rural populations. For administrative purposes, the county includes six sub-

counties and 30 wards, with 146 health care facilities including 90 primary health care facilities [19, 24, 25]. The governance structure of health care services in Kenya is organized into three streams: *partnership*, *governance* and *stewardship* [2]. Partnerships involve stakeholders in health related issues in a given jurisdiction; governance involves different community members in decision making and the management of health facilities; stewardship relates to people working in the government health ministry itself and their participation in decision making. These are set out in Fig 1.

## Study population

Participants were purposely selected from stakeholders including NGOs, professional bodies, private service providers, and the media, from health stakeholder forums, and community members of primary health care committees (Fig 1). Five primary health care facilities were selected purposively from different sub-counties, allowing for maximum variation and potential to provide rich information on interactions between facility staff and local communities. The criteria for facility selection included work load, location and potential to obtain information; senior officers of the Department of Health advised on this and facilitated entry of the first author (JS) (Table 2).

## Data collection methods

The main methods used for data collection were: review of documents, including primary source material from health facilities; key informant interviews; and observations, as described below. Multiple methods were used to ensure the reliability and validity of the research findings [26, 27]. All data were collected by JS with assistance from two trained research assistants who took notes and transcribed audio recordings. JS also took field notes and maintained a reflective journal to aid in the interpretation of meanings. The process was overseen by the supervisors (co-authors LM and MN).

**Review of documents.** The policy documents reviewed included the Uasin Gishu Strategic Plan 2013–2018 [19], National Health Sector Strategic Plan 2014–2018 [2], Uasin Gishu County Project Implementation Plan, Kenya Health Policy Plan 2015–2030 [25] and the Kenya Constitution of 2010 [28]. In reviewing these documents, the authors examined governance, community participation, and stakeholder involvement. Minutes of the proceedings of the health facility committees were also reviewed for the period from 2014 to April 2017, following government elections in 2013 and the subsequent implementation of policies concerning collaborative governance in 2014. Committee minutes provided a basis for triangulating data from interviews and observations.

**Interviews.** A total of 60 participants participated in observations, and of these, 36 were interviewed. They included 25 people who were members of the five facility committees, with three executive members (chairperson, treasurer, secretary) and two other members from each case study facility committee. We also interviewed 11 stakeholders representing other institutions interested in health service delivery at the local level. In consultation with the clinical officer or nursing officer in charge of the facility (referred to locally and in this article as 'facility in-charge'), participants were selected purposively to ensure representation of various sections. Prior arrangements were made before the interviews were carried out at the facilities. Each interview took around 30 minutes. A question guide was used to aid data collection, developed by the authors based on the literature on governance. The main variables under study were structures in place such as selection process (democratic, transparent, inclusive) and ability for representation (voice, power) and challenges of collaboration. Example of the questions included: "How were you selected to represent the community?" "Are there any challenges

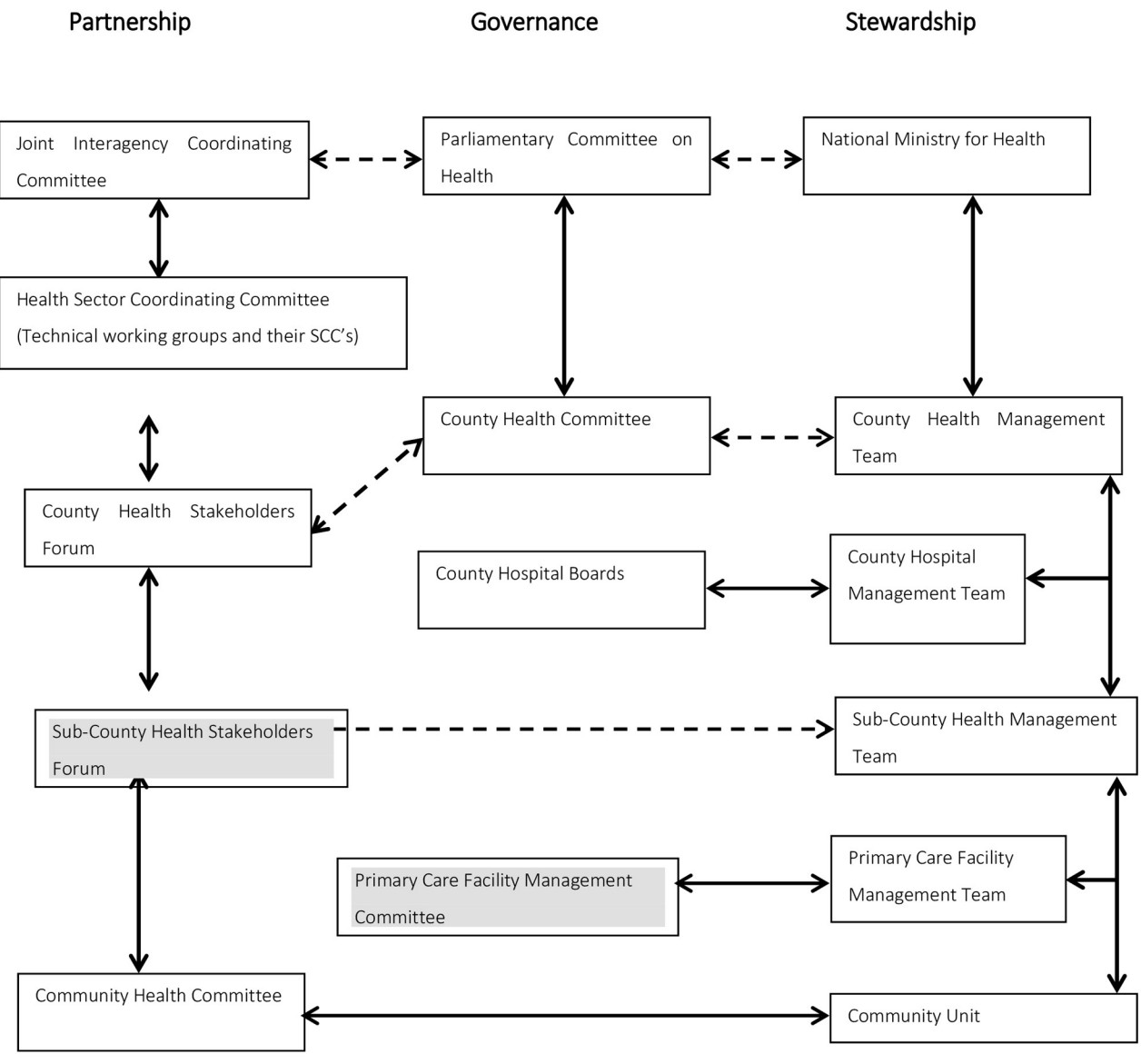

**Fig 1. Governance structure for the Department of Health, in Kenya, national and county governments.** Source: Government of Kenya[1]. Key: Shading shows from where participants are drawn.

that you experience when carrying out your duties?" Interviews were audio recorded to ensure the accuracy of data captured [26].

**Observation.** Using an observation guide, JS observed a total of five meetings (one from each primary health facility) as a non-participant outsider. Observations provided opportunities for further enquiry, understanding and clarification of issues under study [26, 27], such as indications of power differences among committee members that might affect their

**Table 2. Primary health care facility.**

| Primary healthcare facility | 1 | 2 | 3 | 4 | 5 |
|---|---|---|---|---|---|
| Location | Semi-Urban | Urban | Semi-urban | Rural | Rural |
| Catchment population | 39000 | 14932 | 17460 | 10470 | 16200 |
| No. of villages served | 12 | 12 | 10 | 18 | 13 |
| Operations | | | | | |
| Weekdays 8-5pm | | ✓ | | ✓ | |
| Everyday 24 hours a day for maternity and outpatient | ✓ | | ✓ | | ✓ |
| Staffing | | | | | |
| Medical Officer | 0 | 0 | 2 | 0 | 0 |
| Clinical Officer | 6 | 4 | 3 | 1 | 4 |
| Nursing Officer | 12 | 11 | 14 | 3 | 9 |
| Pharmaceutical Officer | 2 | 2 | 2 | 1 | 1 |
| Public Health Officer | 1 | 1 | 1 | 1 | 1 |
| Records Officer | 1 | 1 | 3 | 0 | 1 |
| Watchman/Cleaner | 8 | 3 | 4 | 3 | 5 |
| Record system | | | | | |
| Electronic system | CCC | CCC, OP, Lab, Pharmacy, cash office | None | None | None |
| Paper medical system | OP, MAT, ANC, PNC, CWC, | CWC, ANC, PNC | All paper | All paper | All Paper |

**Key: CCC** Comprehensive care clinic, **OP**–Outpatient clinic, **MAT**–Maternity Clinic, **ANC**- Antenatal Clinic, **PNC**–Postnatal clinic, **CWC**–Child welfare clinic
**Source: Kenya,** Ministry of Health records, Uasin Gishu County, 2017 showing characteristics of primary health care facilities selected as cases

participation in proceedings, the prioritization or presentation of issues, and outcomes [29]. Permission, consent and preparation took place prior to observations.

## Data analysis

Demographic data were entered into Excel and descriptively analysed for frequencies and percentages. All audio recorded interviews were transcribed verbatim into Word, and imported into NVivo12 [27] for analytic purposes. Data were categorized and coded both deductively and inductively for emerging themes as presented in Table 4. Siddique's framework on governance plus other literature provided the lens from which community participation in governance was examined [1, 6, 30, 31]. Siddique proposes examining governance from a multilevel perspective, and includes as attributes of good governance accountability, community participation and consensus orientation, transparency, information, equity and inclusion, efficiency, effectiveness and power. These attributes guided the formulation of the interview schedules. Emerging themes from the data were also coded and presented. Negative or contrary findings were identified and reported. Observations by the primary researcher informed the interpretation of interviews and were discussed among all authors [32–35].

## Reflexivity

JS conducted the research as a Ph.D. student in Public Health, with a nursing, research ethics and public health background. She currently works in an academic institution in health systems management, and had previously worked in various management positions in the health sector. This had the potential to influence the collection of data, including through prestige bias responses in interviews, observer effects, and observer bias in interpreting findings. However, insider status for qualitative researchers has the potential also to strengthen engagement and open discussion through shared knowledge of context and history. Effort was made to set

aside personal biases by presenting participants' perspectives through quotes. Prolonged engagement and understanding of the governance framework aided development of data collection and analysis. LM and MN provided guidance during study design, data collection and analysis.

## Ethics

Ethics approval was granted by Moi University and Moi Teaching and Referral Hospital Institutional Review and Ethics Committee (IREC) 0015931 and the Human Research Ethics Committee (Medical) of University of the Witwatersrand (clearance certificate M170497). Written permission to conduct the study was also given by the Department of Health of Uasin Gishu County. Participants were briefed about the purpose of the study, and those willing were asked to sign an informed consent form for all aspects of the study, including audio recording. Participants were assured of anonymity for any information given. Ethical issues associated with data management such as confidentiality, de-identifying the data, and safe keeping were upheld at all stages.

## Results

### i.) Demographic characteristics

The demographic profiles of the 60 participants representing individuals service providers and community members in collaborative governance are presented in Table 3. Two thirds of the participants were male (39, 65%), with almost half (27, 45%) aged between 30 to 40 years. Fifty percent (30, 50%) had a secondary level of education.

**Table 3. Demographic profile of the participants.**

| Primary Health Care Facility | 1 | 2 | 3 | 4 | 5 | Other stakeholders | Total (n) |
|---|---|---|---|---|---|---|---|
| No. of HFC members | 11 | 9 | 9 | 9 | 11 | 11 | 60 (100%) |
| Gender | | | | | | | |
| ▪ Female | 3 | 4 | 2 | 3 | 4 | 5 | 21 (35%) |
| ▪ Male | 8 | 5 | 7 | 6 | 7 | 6 | 39 (65%) |
| Age bracket | | | | | | | |
| ▪ 30–40 | 4 | 3 | 3 | 4 | 6 | 7 | 27 (45%) |
| ▪ 41–50 | 4 | 3 | 2 | 4 | 4 | 3 | 20 (33%) |
| ▪ 51–60 | 2 | 3 | 4 | 1 | 1 | 1 | 12 (20%) |
| ▪ 61–70 | | 1 | | | | | 1 (2%) |
| Position in HFC | | | | | | | |
| ▪ Executive | 4 | 3 | 4 | 4 | 3 | | 18 (30%) |
| ▪ Member | 6 | 3 | 5 | 5 | 6 | | 25 (42%) |
| Ex-officio | 1 | 1 | 2 | 0 | 2 | | 6 (10%) |
| Others | | | | | | 11 | 11(18%) |
| Education | | | | | | | |
| ▪ Secondary | 6 | 5 | 7 | 6 | 6 | | 30 (50%) |
| ▪ Certificate | 2 | 2 | 0 | 0 | 1 | 1 | 6 (10%) |
| ▪ Diploma | 3 | 2 | 2 | 2 | 3 | 2 | 14 (23%) |
| ▪ Degree | 0 | 2 | 0 | 0 | 0 | 8 | 10 (17%) |
| Public finance management training | 3 | 3 | 3 | 3 | 3 | | 15 (25%) |
| Not trained in public finance management | 8 | 6 | 6 | 6 | 8 | 11 | 45 (75%) |

Source: Research data, 2017.

## ii.) Transparent and democratic election

The election process was generally open, democratic and transparent with community involvement to some extent. The participants and review of documents revealed that after elapse of three years term period, the positions were advertised by respective government officers from the department of health, on the election day there was a big meeting at the facility grounds for the election and there was a document giving the characteristics of potential people who would be chosen, however the attributes were not clearly stipulated.

**a) Advertisement of vacant position.** Office bearers are elected for three-year terms. Towards the expiry of tenure of service, the Office of the County Administrator advertises the need to elect new members through posters and through oral communication with the assistance of government appointed chiefs, community health volunteers and ward administrators. The date of the election is posted on notice boards at health facilities and shops. Announcements are also made on local radio stations (Kass FM and Radio Upendo). In the election held in two of the primary health care facilities during the study period, phone calls, text messages and social media platforms (Facebook and WhatsApp) aided in information transmission. This led interviewees to comment that everyone was aware of the process and subsequent outcomes.

**b) Election day.** On Election Day, at most facilities, meetings were held at the health facility grounds to seek nominations; these were presided over by the area chief, the public health officer and administrators from the Country Department of Health. The meetings were attended by community members, administrative and political leaders, and officers in charge of the health facilities. For the two elections attended by the first author, the meetings commenced with speeches from politicians such as Members of the County Assembly (MCA). Community members were reminded of the election regulations and procedures, and were then divided into groups according to village. Each village nominated one person to act as their representative on the health facility committee. The selected persons were then presented to all participants for their agreement by acclamation to election.

Observations of the two elections by JS showed that attendance was low and not proportionate to the estimated population. Two villages were not represented at all, and in these cases, the chief was told to go and organize the village to select their representative and present the name to the administrators. In addition, generally more men than women attended community meetings. The villagers sat on the grass, men on one side applauding the speeches, women on the other side, their feet outstretched. The county team (all visitors who held public office) who provide leadership and oversight sat on chairs, drawing attention to the power differences between them and community members.

After consensus was reached on potential membership, the names were forwarded to the Chief Executive Officer of Health for endorsement before the elected individuals were able to assume office. If the nominations resulted in a lack of gender balance (usually too few women, who should constitute at least 30% of total membership as per affirmative action goals captured in the constitution) or failed to include members from special groups (youth and disabled people), then the process would be repeated, although in the meetings that the first author attended, people with disabilities were not represented. Sometimes election results were decided behind the scenes, and at least some interviewees were ambivalent about the process and the suitability of candidates, as one interviewee remarked: "The selection is political. They are selected from villages. Also, the people who are young shy away from participating and we end up with people who are not well educated" (male respondent, 51–60 years). JS also observed at one meeting that some people raised concerns that one person who was elected had previously mismanaged money in the community when he was a primary school

committee member. Their concerns were dismissed by the presiding team, and the person was registered as a new committee member; this suggests that community opinion carried little weight with the authorities.

### iii.) Attributes

The Projects Implementation and Management Act (2014) specifies only the requisites of the chairperson and treasurer. The chairperson was required to have previous experience in leading public health community projects and minimum ordinary level education, while the treasurer had to have been trained in financial management. The primary facility in-charge, either the nurse in-charge or clinical officer in-charge, held the position of secretary. Some participants thought that these criteria were ignored when choosing committee members; rather community members selected a person based on their perception that the person had the ability to lead, could unite people, and responded well to community issues: "Trainability of some community members proves a challenge by virtue of their background education, hence some find it difficult to comprehend the training sessions and activities" (male respondent, 30–40 years).

### iv.) HFC meetings and agenda

According to the Projects Implementation and Management Act, a maximum of eight meetings should be held in a financial year, with five members constituting a quorum; an allowance of Ksh. 500 (US $5) per head to cover lunch was set aside for those who attended. The policy document stipulates that the County Executive Committee (CEC) member for health needs to grant permission for extra sitting days to discuss issues related to the day-to-day functioning of the health facility, because of financial implications. In the study period, health facility committees met between seven and 12 times in a financial year, hence slightly more often than stipulated. The required quorum was not reached in seven instances across the facilities, but meetings proceeded. Only one facility publicly posted the year's schedule of health facility committee meetings, ensuring that they were open to public involvement and awareness of meetings. Fig 2 shows the frequency of agenda items at these meetings.

The most common items were financing, including budgeting, payment of contractors, cash withdrawals and change of signatories. Discussions on projects included follow up of ongoing construction at the facility, such as building a maternity wing or a new ward, renovating rooms, fencing, painting and repairs. Other agenda items included the purchase of land adjacent to facilities, purchase of a water tank, quotations for and purchase of an ambulance, and the sale of maize grown on common land adjacent to the facility. At one facility committee meeting, there was an extended discussion on the formulation of Authority to Incur Expenditure (AIE). Observation of the meetings, review of documents, and interviews confirmed this focus of items.

The majority of respondents revealed that there was a recurrent shortage of drugs at all facilities, reflecting community frustration with constant stock outs.

Days like Monday or Tuesday, there are a lot of people. You may find that there are no drugs. From the county, the drugs arrive very late. The patient may have to go and buy prescribed drugs yet not all of them may afford. Sometimes we go and borrow drugs in the neighbouring facilities. We try and maintain our facility though it reaches a breaking point. If it is an injection, you have to go and buy before you are injected. It is cumbersome (Female respondent, 30–40 committee member).

The procurement process is long and tedious, like the jargon, but we hope to get used to this by way of training and exposure of the committee members. We have cases of misuse of funds by some of the committee members since they are used to the old ways of management

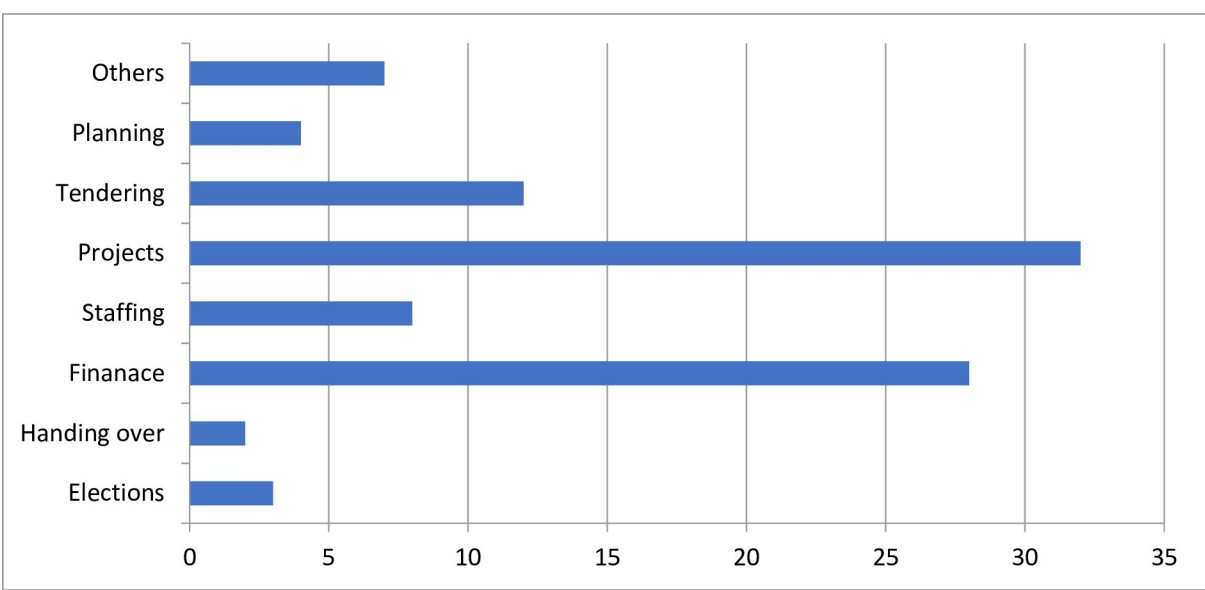

**Fig 2. Frequency with which the issues were discussed at HFC meetings.** Source: Research data, 2017.

of resources including the deliberation that is done by a few people who are not members of the community (male respondent, 30–40 years).

## v.) Practices of powers

The possession and exercise of power varied among members across committees. Community representatives on the health facility committees held power by virtue of their election, but there was little evidence of their utilization of this power in collaborative governance processes. The administrators, Members of the County Assembly, and facility in-charges, on the other hand, exercised considerable power, ran the meetings and dominated discussions.

Community members delegated decision-making power to their representatives in the committee, whether or not they participated in the election of representatives. There were no formal criteria for committee members to represent community priorities, no mechanisms for them to revert to the community in the case of emerging issues or controversy, and no way generally for them to formally consult and communicate with others in their communities. On the other hand, some health facility committee members could exercise power attributed to knowing other more powerful individuals in government and in civil society, for instance, by making direct phone calls to people such as the Governor or County Administrator to raise their concerns.

There was an incident that occurred recently at one of the facilities; someone posted, "can the governor intervene, there is a very long queue at the pharmacy?" Almost immediately, this was acted upon. Another person informed the governor that the facilities were not open yet it was 11am, already way past time. We were able to intervene through this information we received. Other people even tell us that the drugs are out of stock and so we use this information to get more drugs (Male respondent, 30–40 years, leadership position).

In some facilities, the two community representatives, in their capacity as chairperson and treasurer, had total control over all activities, to the frustration of the facility in-charge.

Sometimes members of the committee disagreed about financial commitment and use, and this led to frustration by those in charge of the facilities since dissenting views on finances from the committee members affected both everyday operations and planning processes.

> We have had arrogance from some committee members such that they want to be seen to be the ones controlling the hospitals or the facilities, yet they may not have the leadership qualities or administrative knowledge or management skills. Some community members with regard to their community status, what they are, their social standing or wealth, want to run the projects yet they do not have the capacity causing us to experience a lot of arrogance (Male respondent, 30–40 years, leadership position).

Administrators also enjoyed information power. At times they withheld vital information from the county government concerning the management of health facilities, and were able to do so since they reported directly to the governors' office. They also possessed legitimacy and connection power, mainly to political leaders and the people whose interests they represented. The administrators acted as the overall authority, and consistent with this, wanted to know and do everything, and to exclude others in the process. This caused distress among other community members and even some committee members.

Those in charge of the facilities (nurses and clinical officers), who were secretaries to the committees, had the greatest power, since they were the accounting officers and possessed technical information about the provision of health services, decision making, accounts, general staffing, facility performance and how the government functions. They were also connected to more senior health authorities, and to fellow staff and community members. Sometimes when community representatives wanted to work contrary to regulations, the facility in-charge would assert his or her authority, ass one person explained:

> I tell them what we have to do according to the rules and regulations of the government. I have to make the authority to incur expenses (AIE) clear to them, and make them understand that we have to follow the rules and regulations. (Female respondent, 30–40 years, leadership).

## vi.) Challenges in coordinating collaborative governance

**Unclear clarification of roles and responsibilities.** As described above, the health facility committee is the main mechanism for collaborative governance, as set out in the county's Projects Implementation and Management Act of 2014. There are weaknesses in the Act, however, affecting its implementation, include the lack of: associated regulations and ordinances; information about procedures; processes for scrutiny of elected members; and verification of capacities and the abilities of elected members to hold office and represent community members on the committee. Some health committee members were aware of the Act and its implications; others were not. There were no mechanisms to scrutinise the attributes and capabilities of HFC members, including in relation to integrity, past experience, accountability and ability to manage. The persons in-charge of the facility were automatically secretaries to the committee, as noted above, and so were not vetted for their ability and interest in representing the government and the community. In cases where the in-charge was performing inadequately, the committee tended to function inefficiently and this ultimately affected service delivery.

The Act did not clearly clarify the respective roles of the new positions created after devolution such as 'administrators', and this led to confusion and conflicts. Some administrators insight on the right to approve the budgets and at the same time be signatories of financial

accounts. Some facility in-charges lamented that there was too much intrusion from the committees, even on small procurements, and they queried whose interests were being served in this context. However, one administrator thought otherwise:

> At first there was a lot of opposition because they were not so used to having someone very close to supervise them but they have now come to accept that they have to work under close supervision of the administrator (Female respondent, 30–40 years).

**Conflict of interests.** In awarding tenders, committee members were supposed to be neutral and non-partisan. However, some committee members were involved in tendering processes and interfered with awards. This led to sub-standard jobs and loss of resources, as one participant who held a position in government office observed: "There could be conflicting interests between the community and the administration. There could be something that the board would want to do, but it becomes difficult for the community to benefit because the implementers are selfish people who only consider their interests" (male respondent, 41–50 years).

**Lack of elaborate structures for stakeholder involvement.** The stakeholders included representatives of relief organizations such as the Red Cross, national government, MPs (Member of Parliament) and MCAs (Members of the County Assembly), community-based organizations, and private groups involved in public health service provision. Most respondents thought that these organizations and individuals all contributed to the provision of resources and services, both for the operation of health centres and for committees to function. These included: financial resources, training, infrastructure development like buildings, specialized services like HIV care, sanitation, supportive supervision, computers and electronic medical records, and human resources. Stakeholder committee members were mainly sourced through lobbying and advocacy by stakeholders to help the Ministry of Health and primary care facilities achieve their objectives. Although stakeholders met and engaged with Ministry of Health staff and facility in-charges to discuss and monitor performance, the agenda of a given funder took priority. There were no clear procedures on how the activities of stakeholders such as NGOs or the Constituency Development Fund (CDF) were controlled and monitored. This resulted in poorly controlled entry and exit of partners interested in the provision of health care services. This in turn led to incomplete projects at the facilities, lack of continuity for initiatives, and lack of community ownership. It also enabled the potential duplication of activities, double funding, and the abandonment of incomplete projects.

## Discussion

The study examined the implementation of collaborative governance in primary health care facilities in Uasin Gishu County, Western Kenya, under the policy of devolved governance of 2013. Although there are documented structures to support collaborative governance functioning, the operationalization, functioning processes, roles, and attributes of the office bearers were unclear. This is not uncommon. Other scholars have also noted that community health committees are limited by lack of clearly defined roles and responsibilities [15]. In The Philippines, for example, similar challenges of collaborative governance on local health boards have been documented [36], including conflicts between authorities and other stakeholders in relation to their respective roles, and boards were not always valued by municipal health officers. Some board members felt that they lacked influence to make health plans. This frustrated efforts of collaborative governance and intimidated community members.

In the study setting, the involvement of community members as consumers and providers in decisions about service were central to the provision of health care. This is consistent with the recommendation of the World Health Organization that community members have a right and duty to participate in planning and the implementation of their own health services [37]. This remains a focus for the attainment of universal health coverage. The legal documents required representation on the committee from women, youths and persons with disabilities, reflecting the 2010 Constitution of Kenya (Art. 27), which specifies that vulnerable, previously discriminated and marginalized groups be included in governance in all public sector committees [28, 38–40]. However, at times several rounds of nominations to health facility committees were necessary to ensure such representation. Decision makers found involvement of members of marginalized groups to be minimal, as we have reported elsewhere [23]. This suggests that more effort needs to be made to enhance the inclusion of such groups. Further, at the time of this study, HFC membership had been extended to a new group of members—administrators representing the county government—whose roles and responsibilities were not clearly outlined.

At county level, HFCs were established under the Projects Management Act 2014, but the act provided contradictory information on the technical skills and requirements for individuals to be committee members. This included the specification that an O level was the minimum education qualification, but also that nomination was open to anyone without secondary education with good social repute and past experience in managing community institutions. The capacity of members of committees to actively participate in the deliberation of issues requires literacy, since members need to be able to read supporting documentation, and hence the effectiveness of committees can be compromised if individuals cannot make informed decisions, or comprehend and critically analyse health care and financial issues. In a study conducted in Kilifi, Kenya, Goodman and colleagues indicated that 34.1% of committee members had not attended school, while 34.2% and 41.3% could not read Kiswahili and English respectively [38]. The authors argued that the high levels of illiteracy among HFC members hampered their decision making abilities [38]. In this present study, in contrast, only one member of all HFCs had primary level schooling as the highest educational qualification, and all others had a least some secondary schooling; in theory therefore committee members were able to participate in an informed manner.

The study findings show that, at face value, the election processes were conducted in an open and transparent manner that enabled community participation. Localized word of mouth and posters were used for communication. However, although these were the most common modes of communication, they appear to have been inadequate and ineffective to encourage community members to participate since turnout was always less than half of the expected population. This suggests lack of interest in HFC and/or the ineffectiveness of communication, and concurs with a study conducted by Odini [41, 42] in Western Kenya, in which people in rural areas appeared to lack interest in information aimed at fostering their participation in community service provision projects [40, 41].

As described in (Table 4), the executive members of all HFCs (chairperson, secretary and treasurer) were trained on how to run committees and on the Public Finance Management Act, and these trainings empowered them. In Kilifi, Goodman and colleagues documented that members who were more educated and trained on health facility management exhibited more authority and tended to intimidate the less informed members of the committees [38]. Lodenstein and colleagues, in exploring the role of health facility committees in Benin, Democratic Republic of Congo and Guinea [31], noted that the operations of HFCs were individualized and not systematic, with minimal or no community consultation. This led to the exclusion of marginalized voices, inability to provide feedback to communities, and difficulties

**Table 4. Codes, categories and themes of election process and challenges of implementing collaborative governance.**

| Code | Sub theme | Theme |
|---|---|---|
| • Code 1. Elected by community | Active advertisement of vacant position | Open, democratic, transparent, inclusive process with community involvement. |
| • Code 2. Communication was through community leaders | Representation of different groups | |
| • Code 3. Distribution of posters | | |
| • Code 1. Elected to represent the youth | | |
| • Code 2. As in charge of facility automatically makes me the secretary | | |
| • Code 3. Elected as chairman | | |
| • Code 4. Women represented gender category | | |
| • Code 5. There was a pastor | | |
| • Code 6. The women represented the gender category | | |
| • Code 7. People propose names | | |
| • Code 8. People voted. . .. with most votes took the day | | |
| • Code 9. We were selected through proposal and seconding | | |
| • Code 10. Was elected to represent people with disabilities | | |
| • Code 11. We were selected through proposing and seconding by show of hands | | |
| • Chairman must have a degree | Available criteria for different responsibilities though not understood by all | Attributes |
| • Vice- chairman must have a degree | | |
| • Code 1. Project treasurer shall have knowledge in finance and administration matters | | |
| • Code 2. People propose names | | |
| • Code 3. As in charge of the facility I automatically became the secretary | | |
| • Code 4. No established criteria | | |
| • Code 5. . . kind of services such a person can offer, behaviour, ability to bring people together and how they respond when there is a problem. | | |
| • Code 1. As a civil servant, I am answerable and not the committee because they are just community members and have no roles as civil servants. | Unclear clarification of roles and responsibilities: It can be deduced that the community members do not clearly know their roles and responsibilities | Challenges |
| • Code 2. . . to make decisions contrary to their wishes | | |
| • Code 3. to stand my ground because I will be answerable. | | |
| • Code 4. have to do according to the rules and regulations of the government | | |
| • Code 5. role is a bit passive without their signatures there is nothing that will go through. | | |
| • Code 6. Other places, the ward administrators are beginning to play a major role and, in some accounts, only the two of you will sign. This is probably political malice. | Other interests | |
| | Conflict of interest | |
| • Code 1. Before the budget that you make as a committee is approved, the . . . . . . should stamp. He was involved. That was a rule some months back from the office of the . . .. I don't know what happened. There might have been some corruption issues. We just do what we are told. | Lack of structures for appropriate stakeholder involvement | |
| • Code 2 they should not intrude so much as to being mandatory signatories, or the daily running of the facility. Maybe they should be involved in county development projects worth millions | | |
| • Code 6. . . ..This creates a loophole it is important to have legislature correct | | |
| • Code 6 Vested interests, arrogance, representation through prerecessions is hard to come by, misuse of funds by some committee members, community members do not have capacity, lack of resources, delayed funding | | |

working with formal administrative structures to articulate health facility issues [31]. In the study reported in this article, there were differences in practices of power and an imbalance of participation among committee members. Facility in-charges, for example, held a lot of power, although not all exercised this effectively. Since executive members were trained on the Public Finance Management Act, they possessed technical information on running facilities and displayed greater power in decision making than other committee members. Choonara [37], studying the importance of on-the-job training and leadership development in South Africa, also found that training helped develop leadership and financial management skills in primary health facility teams. Ansell and Gash [43] have argued that power imbalances are a common problem among stakeholders in collaborative governance, and that measures should be put in place to empower those who are weaker and poorly represented [29, 43–45]. As stated earlier, leadership is critical in facilitating broad and active participation, ensuring broad influence and control of group dynamics [43].

The United Nations' SDG Goal 17 encourages the engagement of partners. The policy documents which support the health care framework envision attaining improved health care through stakeholder collaboration [46, 51, 52]. Stakeholder participation in decision-making and the implementation of health care services was emphasized by respondents as important in collaborative governance at multiple levels of the health system [33, 47–52]. While stakeholders played a positive role in primary health care provision, there were no clear mechanisms for controlling or regulating their entry, operations and exit [50]. In Uasin Gishu County, stakeholder engagement was the direct responsibility of the Chief Executive Officer at the county level [10], and therefore this officer should have been able to put structures in place for effective collaborative process.

## Strengths and limitations of the study

The study adopted case study methods which provided opportunities to explore collaborative governance as implemented in Uasin Gishu County. Multiple methods were used in data collection to strengthening the reliability of findings. The use of a governance framework to aid in the collection, analysis and interpretation of data assisted in focusing the study, so ensuring that all dimensions of interest were covered. The main study limitation is that the findings may not be generalizable, but they do provide a basis for further studies and highlight areas for further in-depth analysis.

## Conclusion

The facility committee members contributed to discussions in relation to construction projects as opposed to the day-to-day functioning of the facility, and meetings were most frequent when project funds were discussed. Committee members with political connections and retired government workers had most influence and power. Lack of clarification of roles and responsibilities, conflict of interests, elaborate structures for stakeholder involvement, and variable competence of health facility committee members were the major challenges to enable collaborative governance. There was weakness in the inclusion of members of marginalized groups, and lack of clarity around processes involving stakeholders, the duplication of projects, and other difficulties associated with the provision, management and sustainability of activities. This suggests that structures for stakeholder involvement and health facility committee functioning need to be strengthened to support the effective implementation of collaborative governance.

## Supporting information

**S1 Checklist. Personal guidelines for documents review on community engagement.**
(DOCX)

**S1 File. Respondents at collaborative level.**
(DOCX)

## Author Contributions

**Conceptualization:** Jackline Sitienei, Lenore Manderson, Mabel Nangami.

**Data curation:** Jackline Sitienei, Lenore Manderson, Mabel Nangami.

**Formal analysis:** Jackline Sitienei, Lenore Manderson, Mabel Nangami.

**Investigation:** Mabel Nangami.

**Methodology:** Lenore Manderson, Mabel Nangami.

**Supervision:** Lenore Manderson, Mabel Nangami.

**Writing – original draft:** Jackline Sitienei, Lenore Manderson, Mabel Nangami.

**Writing – review & editing:** Jackline Sitienei, Lenore Manderson, Mabel Nangami.

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
