## [Decision Letter · Decision Letter 0]

6 May 2020

PONE-D-20-06799

Community participation in collaborative governance of primary health care facilities in Uasin Gishu County Kenya

PLOS ONE

Dear Mrs Sitienei,

Thank you for submitting your manuscript to PLOS ONE. After careful consideration, we feel that it has merit but does not fully meet PLOS ONE’s publication criteria as it currently stands. Therefore, we invite you to submit a revised version of the manuscript that addresses the points raised during the review process.

The reviewers have requested some additions and revisions, particularly regarding clarifications on the methodological aspects of the study, further improvements to the clarity of the written language. In addition to the items raised by the reviewers, please address the following points:

Introduction in abstract section is too long.It is need to more clarify multiple case study methodology in method section.How you ensured your study rigor and trustworthiness?Study objective was not fully transparent and specific. So this issue leads to misclassification of results. Please clarify study objectives and then based on this clarification classify results section in case of facilitators, barriers, ….How you used both deductive and inductive analysis for emerging themes, and how you classified themes? This section needs more clarification.You started result section with election process, you need to classify result based on your study objectives. So this section may be referring to current process of collaborative governance and its process.Please provide thematic table or theme tree, so this enables reader to find themes and sub-themes easily and the relation between themes.

We would appreciate receiving your revised manuscript by Jun 20 2020 11:59PM. To enhance the reproducibility of your results, we recommend that if applicable you deposit your laboratory protocols in protocols.io, where a protocol can be assigned its own identifier (DOI) such that it can be cited independently in the future. For instructions see: http://journals.plos.org/plosone/s/submission-guidelines#loc-laboratory-protocols

We look forward to receiving your revised manuscript.

Kind regards,

Kamal Gholipour, PhD

Academic Editor

PLOS ONE

2. Please provide the institutional email address of the Corresponding Author.

3. Please include additional information regarding the observation and interview guidelines used in the study and ensure that you have provided sufficient details that others could replicate the analyses. For instance, if you developed observation and interview guidelines as part of this study and it is not under a copyright more restrictive than CC-BY, please include a copy, in both the original language and English, as Supporting Information.

4. Please provide additional details regarding participant consent. In the ethics statement in the Methods and online submission information, please ensure that you have specified whether consent was informed.

5. Please include your tables as part of your main manuscript and remove the individual files. Please note that supplementary tables (should remain/ be uploaded) as separate "supporting information" files

6. Please amend either the title on the online submission form (via Edit Submission) or the title in the manuscript so that they are identical.

7. Thank you for stating the following in your Competing Interests section: "none declared"

8. We note that you have indicated that data from this study are available upon request. PLOS only allows data to be available upon request if there are legal or ethical restrictions on sharing data publicly. For information on unacceptable data access restrictions, please see http://journals.plos.org/plosone/s/data-availability#loc-unacceptable-data-access-restrictions.

9. Your ethics statement must appear in the Methods section of your manuscript. If your ethics statement is written in any section besides the Methods, please move it to the Methods section and delete it from any other section. Please also ensure that your ethics statement is included in your manuscript, as the ethics section of your online submission will not be published alongside your manuscript.

Reviewers' comments:

Reviewer's Responses to Questions

**Comments to the Author**

1. Is the manuscript technically sound, and do the data support the conclusions?

Reviewer #1: Partly

Reviewer #2: Yes

2. Has the statistical analysis been performed appropriately and rigorously? 

Reviewer #1: Yes

Reviewer #2: I Don't Know

3. Have the authors made all data underlying the findings in their manuscript fully available?

Reviewer #1: No

Reviewer #2: No

4. Is the manuscript presented in an intelligible fashion and written in standard English?

Reviewer #1: Yes

Reviewer #2: Yes

5. Review Comments to the Author

Reviewer #1: The author utilized a multiple case study methodology, selected five primary health care facilities purposively. Examined the implementation of collaborative governance in primary health care facilities in Uasin Gishu County, Western Kenya, under the policy of devolved governance of 2013. However, what is the significance of the research? How to choose the interviews to ensure representative？Some problems in the citation of references, such as 16-20 17 21，and also a little outmoded.

Reviewer #2: Authors worked on an interesting topic. However, In my opinion this paper needs an extensive modifications by the authors.

General comment: Make a consistent writing style in the whole article. In introduction section you started all paragraphs, leaving some space as contrary in other sections. Focus on editing and some minor grammatical mistakes in the reviewed version.

Introduction:

The first line (introductory) seems to have no link with the next sentences in first paragraph. Modify the first few lines to make a clear introduction of the topic. In addition, I felt lack of details about the significance of community role and participation in primary healthcare. Therefore, the authors may add few lines. Don't mention the place of tables like this: TABLE ONE ABOUT HERE..........it creates confusion. We already know that you placed the tables at the end. In the third paragraph the authors wrote........In 2010, Kenya initiated.....before this line, you can add few lines about the primary healthcare system in Kenya as this is the topic you focused in your study. Make a consistent style throughout the paper. In introduction section in the last lines..........."we examine in-depth the implementation of ......." use the past tense tense as you did in the methods section. Also review this sentence to make it more comprehensible and structured.

Methods:

In study setting, you wrote "This county was selected purposively for its strategic location in the Western region of Kenya, in providing health services"...please mention why western region is important in this regard?

In interview section briefly describe the topic of the question you asked during interviews. Clearly mention the variables of your study in the methods section.

Conclusion: Add some recommending remarks for other countries based on your study.

6. PLOS authors have the option to publish the peer review history of their article (what does this mean?). If published, this will include your full peer review and any attached files.

Reviewer #1: No

Reviewer #2: No

---

## [Author Response · Author response to Decision Letter 0]

7 Aug 2020

PONE-D-20-06799

Community participation in the collaborative governance of primary health care facilities in Uasin Gishu County Kenya

PLOS ONE

A: Responses to the Academic Editor and concerns addressed

1. Introduction in abstract section is too long. 

Introduction in the abstract edited to reduce

length; abstract is now 316 words including subtitles

2. It is need to more clarify multiple case study methodology in method section. Clarified in methods section (page 8). We have included details of study design, study population, methods and management and analysis of data, pp. 7-10

3. How you ensured your study rigor and trustworthiness? We have included a paragraph on reflexivity and research rigor, page 14-15

4. Study objective was not fully transparent and specific. So this issue leads to misclassification of results. Please clarify study objectives and then based on this clarification classify results section in case of facilitators, barriers, …. We examined the implementation of collaborative governance with an interest in the selection process, roles and responsibilities of members, and challenges faced by participants. This is because good governance requires that process should be transparent and participative to allow for accountability, with the goal of effective performance.

5. How you used both deductive and inductive analysis for emerging themes, and how you classified themes? This section needs more clarification. Siddique’s framework on governance plus other literature provided the lens from which community participation in governance was examined. This posits that the attributes of governance include: accountability, transparency, participation, consensus orientation, the provision of information, and ensuring inclusion, regulation and oversight. Emerging themes were also coded and presented through thick description 

6. You started result section with election process, you need to classify result based on your study objectives. So this section may be referring to current process of collaborative governance and its process. The study objectives and results are now consistent. The focus of this article is on the implementation of collaborative governance at primary health care facilities, looking at the structures in place for operations such as the selection processes for membership of the committee, the demarcation and undertaking of roles and responsibilities, and the challenges faced in Uasin Gishu County. In addition, we address the exercise of power, at the time of the election and in proceedings of meetings. 

7. Please provide thematic table or theme tree, so this enables reader to find themes and sub-themes easily and the relation between themes. A thematic table has been provided as supporting information as Table 3 page 14-15

8. Please provide the institutional email address of the Corresponding Author. This has been provided in the manuscript

9. Please include additional information regarding the observation and interview guidelines used in the study and ensure that you have provided sufficient details that others could replicate the analyses. For instance, if you developed observation and interview guidelines as part of this study and it is not under a copyright more restrictive than CC-BY, please include a copy, in both the original language and English, as Supporting Information. Interview and observation guides were developed by the first author in collaboration with the other authors who are the supervisors. The questions were informed by literature review on governance. These instruments are not under any restrictions and are provided as supporting information for readers

10. Please provide additional details regarding participant consent. In the ethics statement in the Methods and online submission information, please ensure that you have specified whether consent was informed. Ethics (Page 15) and online submission ethics statements are provided

11. Please include your tables as part of your main manuscript and remove the individual files. Please note that supplementary tables (should remain/ be uploaded) as separate "supporting information" files Tables have been included as part of the main manuscript

12. Please amend either the title on the online submission form (via Edit Submission) or the title in the manuscript so that they are identical. The title on the online submission has been amended

13. Thank you for stating the following in your Competing Interests section: "none declared” Please complete your Competing Interests on the online submission form to state any Competing Interests. If you have no competing interests, please state "The authors have declared that no competing interests exist.", as detailed online in our guide for authors at http://journals.plos.org/plosone/s/submit-now. This information should be included in your cover letter; we will change the online submission form on your behalf. Please know it is PLOS ONE policy for corresponding authors to declare, on behalf of all authors, all potential competing interests for the purposes of transparency. PLOS defines a competing interest as anything that interferes with, or could reasonably be perceived as interfering with, the full and objective presentation, peer review, editorial decision-making, or publication of research or non-research articles submitted to one of the journals. Competing interests can be financial or non-financial, professional, or personal. Competing interests can arise in relationship to an organization or another person. Please follow this link to our website for more details on competing interests: http://journals.plos.org/plosone/s/competing-interests

The authors declare that no competing interests exist 

14. We note that you have indicated that data from this study are available upon request. PLOS only allows data to be available upon request if there are legal or ethical restrictions on sharing data publicly. For information on unacceptable data access restrictions, please see http://journals.plos.org/plosone/s/data-availability#loc-unacceptable-data-access-restrictions.

Data can be available upon request, as explained in the cover letter. The restrictions relate to anonymity of the facilities and the requirement that any new investigator wishing to access data must apply for ethics clearance. I have elaborated this in the cover letter.

B. In your revised cover letter, please address the following prompts: We have addressed these issues in the cover letter but summaries our responses also below

15. If there are ethical or legal restrictions on sharing a de-identified data set, please explain them in detail (e.g., data contain potentially identifying or sensitive patient information) and who has imposed them (e.g., an ethics committee). Please also provide contact information for a data access committee, ethics committee, or other institutional body to which data requests may be sent. Data can be provided to the editor for the purposes of transparency of the study if need be. However, as explained in the cover letter, governance is a sensitive topic and qualitative data is difficult to de-identify the data completely. Data from this study can be availed through approving ethics committee -Institutional ethics and research Committee (IREC) Moi teaching and referral Hospital (MTRH) and Moi University, College of health Sciences. To access the data one can, write to the board chair, IREC, P.O. Box 3-30100, Eldoret, Kenya. Email: irecmtrh@gmail.com or contact@irec.or.ke. Contact person is Ms Cathrine Okwiri. 

16. If there are no restrictions, please upload the minimal anonymized data set necessary to replicate your study findings as either Supporting Information files or to a stable, public repository and provide us with the relevant URLs, DOIs, or accession numbers. Please see http://www.bmj.com/content/340/bmj.c181.long for guidelines on how to de-identify and prepare clinical data for publication. For a list of acceptable repositories, please see http://journals.plos.org/plosone/s/data-availability#loc-recommended-repositories.

For reasons set out above, data cannot be uploaded and made publicly available.

We will update your Data Availability statement on your behalf to reflect the information you provide. Thank you

17. Your ethics statement must appear in the Methods section of your manuscript. If your ethics statement is written in any section besides the Methods, please move it to the Methods section and delete it from any other section. Please also ensure that your ethics statement is included in your manuscript, as the ethics section of your online submission will not be published alongside your manuscript. Ethics statement has been revised and included in the manuscript (Page 15)

Reviewer's Responses to Questions

C. Comments to the Author

18. Is the manuscript technically sound, and do the data support the conclusions?

Reviewer #1: Partly

Reviewer #2: Yes 

19. Has the statistical analysis been performed appropriately and rigorously?

Reviewer #1: Yes

Reviewer #2: I Don't Know 

20. Have the authors made all data underlying the findings in their manuscript fully available?

Reviewer #1: No

Reviewer #2: No 

21. Is the manuscript presented in an intelligible fashion and written in standard English?

Reviewer #1: Yes

Reviewer #2: Yes 

Reviewer Comments to the Author

24. Reviewer #1: The author utilized a multiple case study methodology, selected five primary health care facilities purposively. Examined the implementation of collaborative governance in primary health care facilities in Uasin Gishu County, Western Kenya, under the policy of devolved governance of 2013. However, what is the significance of the research? How to choose the interviews to ensure representative？ Some problems in the citation of references, such as 16-20 17 21，and also a little outmoded. 

25. Reviewer #2: Authors worked on an interesting topic. However, In my opinion this paper needs an extensive modifications by the authors.

General comment: Make a consistent writing style in the whole article. In introduction section you started all paragraphs, leaving some space as contrary in other sections. Focus on editing and some minor grammatical mistakes in the reviewed version.

Introduction:

The first line (introductory) seems to have no link with the next sentences in first paragraph. Modify the first few lines to make a clear introduction of the topic. In addition, I felt lack of details about the significance of community role and participation in primary healthcare. Therefore, the authors may add few lines. Don't mention the place of tables like this: TABLE ONE ABOUT HERE..........it creates confusion. We already know that you placed the tables at the end. In the third paragraph the authors wrote........In 2010, Kenya initiated.....before this line, you can add few lines about the primary healthcare system in Kenya as this is the topic you focused in your study. Make a consistent style throughout the paper. In introduction section in the last lines..........."we examine in-depth the implementation of ......." use the past tense as you did in the methods section. Also review this sentence to make it more comprehensible and structured.

Methods:

In study setting, you wrote "This county was selected purposively for its strategic location in the Western region of Kenya, in providing health services"...please mention why western region is important in this regard?

In interview section briefly describe the topic of the question you asked during interviews. Clearly mention the variables of your study in the methods section.

Conclusion: Add some recommending remarks for other countries based on your study. Significance of the study has been provided on Page 7, and this is reiterated in the background and in the discussion. We believe that the article will help inform strategies for the more effective implementation of community governance and this should ultimately strengthen social accountability, health outcomes and development

Interviewees were selected to ensure representation, as explained on Page 12-13 of the manuscript. Citation has been revised accordingly.

The paper has been revised extensively including the introduction.

Clauses on the tables have been removed 

A paragraph has been inserted introducing primary health care in Kenya (Page 4)

The reason for the selection of Uasin Gishu County for the study has been provided (Page 10)

Variables under study are explained on page 13

26. PLOS authors have the option to publish the peer review history of their article. If published, this will include your full peer review and any attached files.

Do you want your identity to be public for this peer review? For information about this choice, including consent withdrawal, please see our Privacy Policy.

Reviewer #1: No

Reviewer #2: No

While revising your submission, please upload your figure files to the Preflight Analysis and Conversion Engine (PACE) digital diagnostic tool, https://pacev2.apexcovantage.com/. PACE helps ensure that figures meet PLOS requirements. To use PACE, you must first register as a user. Registration is free. Then, login and navigate to the UPLOAD tab, where you will find detailed instructions on how to use the tool. If you encounter any issues or have any questions when using PACE, please email us at figures@plos.org. Please note that Supporting Information files do not need this step. This is relevant to the reviewers and not to the authors of this manuscript.

---

## [Decision Letter · Decision Letter 1]

5 Oct 2020

PONE-D-20-06799R1

Community participation in the collaborative governance of primary health care facilities, Uasin Gishu County. Kenya

PLOS ONE

Dear Dr. Sitienei,

Thank you for submitting your manuscript to PLOS ONE. After careful consideration, we feel that it has merit but does not fully meet PLOS ONE’s publication criteria as it currently stands. Therefore, we invite you to submit a revised version of the manuscript that addresses the points raised during the review process.

The reviewers find the work of merit but we have requested some additions and revisions, please address the following points:

Please correct and complete the title of table 3 as informative format. Thematic table is part of your result not method section, and must be presented in result section. Also, it appears correct table as right to left format (code, category and theme).

The coding process in its current format is not satisfactory. Codes in some case raised as quotes and in some cases as codes. categories and themes were not related logically and conceptually. You can find a useful structure to coding process in following references:

de Casterlé BD, Gastmans C, Bryon E, Denier Y. QUAGOL: A guide for qualitative data analysis. International journal of nursing studies. 2012 Mar 1;49(3):360-71.Graneheim UH, Lundman B. Qualitative content analysis in nursing research: concepts, procedures and measures to achieve trustworthiness. Nurse education today. 2004 Feb 1;24(2):105-12.Iezadi, S., Tabrizi, J.S., Ghiasi, A. et al. Improvement of the quality payment program by improving data reporting process: an action research. BMC Health Serv Res 18, 692 (2018). https://doi.org/10.1186/s12913-018-3472-4

In order to provide a more complete information to our readers on the topic, we would like to emphasize the importance to cross referencing very recent material on the same topic published in "PLoS ONE ". Therefore, it would be highly appreciated if you would check the contents published in the last two years of "PLoS ONE" (https://journals.plos.org/plosone/) and add all material relevant to your article to the reference list.

We look forward to receiving your revised manuscript.

Kind regards,

Kamal Gholipour, PhD

Academic Editor

PLOS ONE

Reviewers' comments:

Reviewer's Responses to Questions

**Comments to the Author**

1. If the authors have adequately addressed your comments raised in a previous round of review and you feel that this manuscript is now acceptable for publication, you may indicate that here to bypass the “Comments to the Author” section, enter your conflict of interest statement in the “Confidential to Editor” section, and submit your "Accept" recommendation.

Reviewer #1: All comments have been addressed

Reviewer #2: All comments have been addressed

2. Is the manuscript technically sound, and do the data support the conclusions?

Reviewer #1: Yes

Reviewer #2: Yes

3. Has the statistical analysis been performed appropriately and rigorously? 

Reviewer #1: Yes

Reviewer #2: I Don't Know

4. Have the authors made all data underlying the findings in their manuscript fully available?

Reviewer #1: Yes

Reviewer #2: No

5. Is the manuscript presented in an intelligible fashion and written in standard English?

Reviewer #1: Yes

Reviewer #2: Yes

6. Review Comments to the Author

Reviewer #1: Thanks for the revised manuscript. This manuscript revision is in a sufficiently mature form and in my judgement fully acceptable for publishing without further changes.

Reviewer #2: No further comments from my side.Authors did their best to address all comments.However, regarding statistical analysis other reviewers comments are important to respond.

7. PLOS authors have the option to publish the peer review history of their article (what does this mean?). If published, this will include your full peer review and any attached files.

Reviewer #1: No

Reviewer #2: No

---

## [Author Response · Author response to Decision Letter 1]

4 Dec 2020

I have attached a table to response to reviewers. To the best of my ability, i have addressed the concerns raised.

Thank you for review and concerns raised. This will go along way in enriching our manuscript.

---

## [Editor Report · Decision Letter 2]

21 Dec 2020

PONE-D-20-06799R2

Community participation in the collaborative governance of primary health care facilities, Uasin Gishu County. Kenya

PLOS ONE

Dear Dr. Sitienei,

Thank you for submitting your manuscript to PLOS ONE. After careful consideration, we feel that it has merit but does not fully meet PLOS ONE’s publication criteria as it currently stands. Therefore, we invite you to submit a revised version of the manuscript that addresses the points raised during the review process.

The authors did a nice job addressing comments and suggestions. However, I have a few minor additional comments.

In table 4 use bullets to shows each code. Also affiliate every code to each sub theme (replace with category) and every sub theme to each theme. Use same wording in table as text to represent codes, sub theme and themes.

Code

sub theme

theme

Code 1Code 2Code 3

sub theme

theme

Code 1Code 2Code 3

sub theme

We look forward to receiving your revised manuscript.

Kind regards,

Kamal Gholipour, PhD

Academic Editor

PLOS ONE

---

## [Author Response · Author response to Decision Letter 2]

4 Feb 2021

Dear Academic Editor Kamal Gholipour, Editor PLOS ONE,

Thank you for the concerns raised as relates to table 4.

We have addressed them and revised the table as per your guidance in page 24-26, hopefully it should meet PLOS one standards for publication.

Thank you

Regards

Jackline Sitienei

On behalf of the authors

---

## [Editor Report · Decision Letter 3]

9 Mar 2021

Community participation in the collaborative governance of primary health care facilities, Uasin Gishu County. Kenya

PONE-D-20-06799R3

Dear Dr. Sitienei,

We’re pleased to inform you that your manuscript has been judged scientifically suitable for publication and will be formally accepted for publication once it meets all outstanding technical requirements.

Kind regards,

Kamal Gholipour, PhD

Academic Editor

PLOS ONE
---

## [Editor Report · Acceptance letter]

22 Mar 2021

PONE-D-20-06799R3 

Community participation in the collaborative governance of primary health care facilities, Uasin Gishu County. Kenya 

Dear Dr. Sitienei:

I'm pleased to inform you that your manuscript has been deemed suitable for publication in PLOS ONE. Congratulations! Your manuscript is now with our production department. 

Kind regards, 

on behalf of

Dr. Kamal Gholipour 

Academic Editor

PLOS ONE